# Causative Pathogens Do Not Differ between Early, Delayed or Late Fracture-Related Infections

**DOI:** 10.3390/antibiotics11070943

**Published:** 2022-07-14

**Authors:** Ruth A. Corrigan, Jonathan Sliepen, Maria Dudareva, Frank F. A. IJpma, Geertje Govaert, Bridget L. Atkins, Rob Rentenaar, Marjan Wouthuyzen-Bakker, Martin McNally

**Affiliations:** 1Bone Infection Unit, Nuffield Orthopaedic Centre, Oxford University Hospitals, Oxford OX3 7HE, UK; maria.dudareva@ouh.nhs.uk (M.D.); bridget.atkins@ouh.nhs.uk (B.L.A.); martin.mcnally@ouh.nhs.uk (M.M.); 2Nuffield Department of Clinical Laboratory Sciences, Oxford University, John Radcliffe Hospital, Oxford OX3 9DU, UK; 3Department of Trauma Surgery, University Medical Centre Groningen, University of Groningen, 9712 CP Groningen, The Netherlands; j.sliepen@umcg.nl (J.S.); frankijpma@gmail.com (F.F.A.I.); 4Department of Trauma Surgery, University Medical Centre Utrecht, 3584 CX Utrecht, The Netherlands; g.a.m.govaert@umcutrecht.nl; 5Department of Medical Microbiology and Infection Prevention, University Medical Centre Groningen, University of Groningen, 9713 GZ Groningen, The Netherlands; r.j.rentenaar@umcutrecht.nl; 6Department of Medical Microbiology, University Medical Centre Utrecht, 3584 CX Utrecht, The Netherlands; m.wouthuyzen@icloud.com

**Keywords:** fracture-related infection, fracture, infection, microbiology, pathogen, early delayed or late fracture-related infection

## Abstract

Fracture-related infections (FRIs) are classically considered to be early (0–2 weeks), delayed (3–10 weeks) or late (>10 weeks) based on hypothesized differences in causative pathogens and biofilm formation. Treatment strategies often reflect this classification, with debridement, antimicrobial therapy and implant retention (DAIR) preferentially reserved for early FRI. This study examined pathogens isolated from FRI to confirm or refute these hypothesized differences in causative pathogens over time. Cases of FRI managed surgically at three centres between 2015–2019 and followed up for at least one year were included. Data were analysed regarding patient demographics, time from injury and pathogens isolated. Patients who underwent DAIR were also analysed separately. In total, 433 FRIs were studied, including 51 early cases (median time from injury of 2 weeks, interquartile range (IQR) of 1–2 weeks), 82 delayed cases (median time from injury of 5 weeks, IQR of 4–8 weeks) and 300 late cases (median time from injury of 112 weeks, IQR of 40–737 weeks). The type of infection was associated with time since injury; early or delayed FRI are most likely to be polymicrobial, whereas late FRIs are more likely to be culture-negative, or monomicrobial. *Staphylococcus aureus* was the most commonly isolated pathogen at all time points; however, we found no evidence that the type of pathogens isolated in early, delayed or late infections were different (*p* = 0.2). More specifically, we found no evidence for more virulent pathogens (*S. aureus*, Gram-negative aerobic bacilli) in early infections and less virulent pathogens (such as coagulase negative staphylococci) in late infections. In summary, decisions on FRI treatment should not assume microbiological differences related to time since injury. From a microbiological perspective, the relevance of classifying FRI by time since injury remains unclear.

## 1. Introduction

Fracture related infection (FRI) is reported to occur in 1–2% of closed fractures managed with internal fixation, but may complicate up to 30% of open fractures [1]. This complication is associated with both increased mortality, morbidity and socioeconomic cost [2], which makes the discovery of effective treatment strategies a priority.

Since the publication of the definition of fracture-related infection (FRI) in 2018 [3], numerous researchers have attempted to clarify and unify optimal treatment, including both the surgical [4] and antimicrobial components [5,6]. In 1986, it was proposed that the time since fracture fixation was a key determinant of FRI management [7], based upon the hypothesized differences in causative pathogens in early (<2 weeks since fixation), delayed (3–10 weeks) and late (>10 weeks) infections. It has been suggested that the formation of bacterial biofilm in late infections renders these infections less amenable to curative treatment with fixation retention [8]. This is thought to be because bacteria in biofilm are less susceptible to systemic antibiotics [9] and therefore must be physically removed along with the infected implant(s) for optimal treatment outcome [10]. This hypothesis underpins current clinical practice; for example, in a recent study, 85.7% of patients with early infection underwent debridement and implant retention versus just 9.8% of patients with late infections (*p* < 0.01) [11]. Other factors must also be considered, including extent of fracture healing (and therefore whether it is possible to remove the implant) [10] and the relative effectiveness of debridement of different implants [12]. It has also been suggested that early and delayed infections are more likely to be caused by more virulent organisms such as *S. aureus* and *Escherichia coli*, whereas late infections are more likely to be caused by less virulent organisms such as coagulase-negative staphylococci [9,13].

However, there is ambiguity over whether these time intervals refer to time since injury [14], time since fixation [4,6,11] or time since symptom onset [8] and which, if any, definition is most clinically relevant. Furthermore, if managed with appropriate surgery, time from injury alone does not affect outcome in FRI [15]. Nor is the outcome of prosthetic joint infections (PJI) worse if caused by so-called difficult-to-treat pathogens [16]—those defined as biofilm-forming but resistant to theorized antibiofilm antibiotics (rifampicin-resistant staphylococci and ciprofloxacin-resistant Gram-negative aerobic bacteria [4,9]). Indeed, increasing antimicrobial resistance worldwide [17] is also reflected in cases of osteomyelitis over time [18], although data regarding the role of antimicrobial resistance in FRI is currently lacking. Whether or not the types of pathogens isolated in FRI are time-dependent remains controversial, with just one study suggesting that there is a difference [11], whereas another more recent study found no difference [19]. Finally, a recent review of FRI management declared these time intervals as somewhat ‘arbitrary’ [10].

This paper sought to clarify if the classification of FRI based on time is justified with regard to hypothesized differences in the causative pathogens isolated. We also analysed the subgroup of patients who underwent debridement, antimicrobial therapy and implant retention (DAIR) as part of their treatment for FRI. We found no differences in the causative pathogens over time in our patient cohort, nor in the patients managed with DAIR. More explicitly, we found no evidence of more virulent organisms in early infections with less virulent species presenting later.

## 2. Results

In total, 433 FRIs were studied, including 51 early cases (median time since injury of 2 weeks, IQR of 1–2 weeks), 82 delayed cases (median time since injury of 5 weeks, IQR of 4–8 weeks) and 300 late cases (median time since injury of 112 weeks, IQR of 40–737 weeks) (Table 1). In total, 140 patients underwent DAIR (median time since injury of 4 weeks, IQR of 2–8 weeks) (Table 2).

### 2.1. Patient Demographics

The median age of patients included in this study was 51 years (IQR 36–62), and this was not significantly different between patients with early, late or delayed infection (*p* = 0.85) (Table 1). Overall, 70% of patients were male and patient sex did not differ with time since injury (*p* = 0.12). For all patients, the median body mass index (BMI) was 27.3 (IQR 23.4–31.0). There was a significant trend towards increasing BMI with time since injury (early median BMI 23.6, delayed median BMI 25.4, late median BMI 27.8, (*p* = 0.004)).

The tibia (and/or fibula) were the most affected bones (237 cases, 54.7%), followed by the femur (94 cases, 21.7%) and then upper limb (52 cases, 12.0%) (Table 1). The location of infections was significantly different depending on the time since injury (*p* < 0.0001). Post hoc analysis indicates this is due to differences in the late infections group: notably fewer pelvic infections compared to both early and delayed infections, and fewer infections of the feet. In addition, there were more ‘other’ bones, including infections of the patella and sternum. Analysing infections of the tibia (and/or fibula), femur and upper limb only, which accounted for 88% of all infections, revealed no significant difference between the groups (*p* = 0.17). 

### 2.2. Type of Infection in Early, Delayed and Late Infections

Infections were classified as culture-negative, monomicrobial or polymicrobial. Overall, most infections were monomicrobial (199, 45.9%). In total, 154 (35.6%) infections were polymicrobial, and the remainder (80, 18.5%) were culture-negative. The type of infection was significantly different between early, delayed and late infections (*p* < 0.0001). Negative cultures were uncommon before 10 weeks but more frequent in late FRIs (4% vs. 24%, *p* < 0.0001). Conversely, only one quarter of late FRIs were polymicrobial compared to over half of both early (59%, *p* = 0.0004) and delayed FRIs (56%, *p* < 0.0001) (Table 1). To summarise, our data suggest that early or delayed FRIs are most likely to be polymicrobial, whereas late FRIs are more likely to be culture-negative, or monomicrobial. Of note, most late culture-negative FRIs were diagnosed by the presence of a sinus (50, 69%). In total, 68 late culture-negative FRIs had histology results available, of which 35 (51%) were positive for infection. 

### 2.3. Pathogens Isolated in Early, Delayed and Late Infections

In total, 637 pathogens were isolated from 433 FRIs. Across all time points, *S. aureus* was the most commonly isolated pathogen (200, 31%). Aerobic Gram-negative bacteria were next most commonly isolated (142, 22%), followed by other staphylococci species (92, 14%). There were no significant differences in the distribution of species of pathogens isolated in early, delayed or late infections (*p* = 0.2). The distribution of pathogens isolated in early, delayed and late infections is represented in Figure 1. 

#### 2.3.1. *Staphylococcus aureus*

Of all pathogens isolated in early, late and delayed infections, *S. aureus* accounted for 24.4%, 29.8% and 33.8%, respectively, and the distribution of *S. aureus* isolates was not statistically significantly associated with time since injury (*p* = 0.19). Taking into account polymicrobial infections, where multiple isolates of *S. aureus* were identified in a single FRI (two late infections), the percentage of cases caused by at least one strain of *S. aureus* was 45.1%, 58.5% and 42.3% for early, delayed and late infections, respectively. 

Finally, only considering infections where pathogens were isolated, i.e., removing culture-negative cases from the analysis, *S. aureus* was isolated in 46.9%, 63.2% and 55.7% of early, delayed and late cases (*p* = 0.20), respectively. In summary, our data do not support the hypothesis that early infections are more likely to be caused by more virulent pathogens, such as *S. aureus*. 

Data regarding methicillin resistance patterns was available for the UK centre data only. In total, 10 out of 81 (12.3%) *S. aureus* isolates found in late infections were methicillin-resistant. 

#### 2.3.2. Other Staphylococci

For other staphylococci (including *S. epidermidis*), these ‘less virulent’ staphylococci accounted for 18.1%, 14.3% and 13.6% of total pathogens isolated in early, delayed and late infections (*p* = 0.54), respectively. Taking into account polymicrobial infections, the percentage of cases in which at least one strain of coagulase negative staphylococci was isolated was 27.5%, 23.2% and 16.3% for early, delayed and late cases (*p* = 0.1), respectively. Finally, considering only culture positive cases, other staphylococci were isolated in 28.6%, 25.0% and 21.5% of early, delayed and late infections (*p* = 0.52), respectively. In summary, our data do not support the hypothesis that these ‘less virulent’ staphylococci are more likely to be isolated in delayed and late infections. 

Data regarding methicillin resistance patterns was available for the UK centre data only. In total, 4 out of 17 (23.5%) *S. epidermidis* isolates in late infections were methicillin-resistant.

#### 2.3.3. Gram-negative Aerobic Bacteria

Gram-negative aerobic bacteria accounted for 21.3%, 21.1% and 23.0% of total pathogens isolated in early, delayed and late infections (*p* = 0.86), respectively. Taking into account polymicrobial infections, the percentage of cases where at least one Gram-negative aerobic bacilli was isolated was 25.5%, 31.7% and 23.0% (*p* = 0.27) for early, delayed and late cases, respectively. Finally, considering only culture-positive cases, Gram-negative aerobic bacteria were isolated in 26.5%, 34.2% and 30.3% of early, delayed and late infections (*p* = 0.65), respectively. In summary, our data do not support the hypothesis that these ‘more virulent’ pathogens are most likely isolated in early infections. 

Data regarding extended spectrum beta lactamase (ESBL) or carbapenemase (CPE) resistance patterns were available for the UK centre data only. In total, 2 out of 72 (2.8%) and 2 out of 72 (2.8%) Gram-negative isolates in late infections were ESBL and CPE, respectively. 

### 2.4. Difficult to Treat’ Infections

Difficult-to-treat (DTT) infections are defined as rifampicin-resistant staphylococci and ciprofloxacin-resistant aerobic Gram-negative bacilli. In total, 9 out of 291 (3.1%) staphylococcal isolates with sensitivity data were rifampicin-resistant (in one case, *S. aureus* was isolated by 16S PCR only), accounting for 2.5%, 0% and 4.4% of staphylococci in early, delayed and late infections, respectively. In total, 13 out of 142 aerobic Gram-negative isolates were ciprofloxacin-resistant, accounting for 0%, 5.9% and 12.5% of Gram-negatives isolated in early, delayed and late infections, respectively. 

### 2.5. Sub-Analysis of Patients Managed with DAIR

One hundred and forty patients underwent DAIR (median time since injury of 4 weeks, range 0–946 weeks) (Table 2). Patients treated for early, delayed and late infections did not differ in terms of age (*p* = 0.63) or sex (*p* = 0.62), though they did differ in terms of BMI (*p* = 0.027). The most commonly affected bone was the tibia and/or fibula (70, 50%), followed by the femur (26, 19%) and pelvis (20, 14%), and this did not differ between early, late or delayed infections (*p* = 0.26). 

For cases of FRI managed by DAIR, most infections were polymicrobial (74, 52.8%) and then monomicrobial (59, 42.1%). Only seven (5%) infections were culture-negative. The type of infection did not differ between early, delayed and late infections managed by DAIR (*p* = 0.52). 

In total, 264 pathogens were isolated from 140 FRI cases managed with DAIR. Across all time points, *S. aureus* was the most commonly isolated pathogen (81, 31%). There was no significant difference in the distribution of species of pathogens isolated in early, delayed or late infections (*p* = 0.56).

## 3. Discussion

This international collaboration is the largest study examining pathogens isolated in FRI. We found no differences in the pathogens isolated in early, delayed or late FRI. Patient demographics, the most commonly affected bones and the most common pathogens were comparable to other published studies [6,10]. For example, in a recent review, Depypere et al. found that *S. aureus* was isolated in 30–42% of FRI compared to 31% in this study [10]. Our findings that infections were most common in males and in lower extremity fractures correlates with known risk factors for FRI [20]. 

### 3.1. Late FRI Are More Likely to Be Culture-Negative

Our data indicate that early or delayed FRI are most likely to be polymicrobial, whereas late FRIs are more likely to be culture-negative, or monomicrobial. This trend is also seen in another published study, albeit with much smaller numbers [19]. Walter et al. found that only 9.4% of FRI cases were culture-negative compared to 19% overall in our study, but all of these culture-negative infections were found in delayed or late FRI [19]. Our culture-negative rate is higher than other published reports [21]; however, this may be due to the high percentage of late infections in our cohort. Of note, a recent study of PJI defined according to consensus definitions found zero cases of culture-negative infections in acute cases versus seventy (4.7%) in chronic infections, which reflects our findings in FRI [22].

Perhaps in late FRI, organisms are more difficult to culture. Patients may have had multiple previous operations or multiple antibiotic courses to try and eliminate infection. Thus, perhaps pathogens are present in smaller numbers or more sporadic distribution than in earlier infections, leading to sampling errors [23]. Similarly, late FRIs may have had any internal fixation devices removed in previous operations, reducing the opportunity to sonicate any metalwork, a process which, when combined with deep tissue sampling in the diagnosis of prosthetic joint infection, is known to increase culture sensitivity [24]. In late infections, organisms may exist in a quiescent phase, analogous to the so-called small colony variants (SCV) of *S. aureus*, with associated fastidious culture requirements [25], making negative cultures more likely. Such SCVs have been isolated in persistent and chronic infections, including osteomyelitis [26] and within osteocytes from clinical cases of PJI [27]. Polymerase chain reaction (PCR) has been shown to be useful to isolate pathogens in culture-negative PJI, though studies looking at its utility in FRI are still lacking [28]. 

### 3.2. Assumptions Regarding Different Pathogens in Early, Delayed and Late FRI Should Not Be Used to Guide Antimicrobial Therapy

We found no association between bacterial species isolated in FRI and time from injury. At all time points, *S. aureus* was the most commonly isolated pathogen, in agreement with other published studies on FRI [6,10,11]. We also found no evidence that bacteria presumed to be more virulent (*S. aureus*, aerobic Gram-negative bacilli) are more likely in early infections and less virulent organisms (coagulase negative staphylococci) are more likely in delayed or late infections. Of note, Kuehl et al. did find that, although *S. aureus* was the most common pathogen at all times, the second most common in early infections were Enterobacteriaceae, compared to coagulase negative staphylococci in late infections. However, this study also found no association between any pathogen and treatment failure, perhaps calling into question the validity of suppositions about pathogen virulence in the context of FRI [11]. 

In our study, numbers are too small to comment on any differences in resistant pathogens (including so-called DTT pathogens). Although increasing antimicrobial resistance has been observed in osteomyelitis over time, and presents treatment challenges [18], such resistance patterns may reflect trends in antimicrobial use [29] and highlight the need for antimicrobial stewardship in orthopaedic infections [30]. Educating prescribers on the consequences of inappropriate antimicrobial use must therefore be a priority [31].

Given the absence of an association between bacterial species, virulence or resistance with time from injury, time from injury should not be used to determine antimicrobial therapy. This is especially important when managing culture-negative infections. We found no evidence that the higher incidence of culture-negative infections in late FRI is due to difference in bacterial species or presumed virulence. Thus, when managing culture-negative infections, clinicians must consider antibiotic regimes active against the diversity of pathogens isolated in all culture-positive infections.

### 3.3. The Role of Biofilm in Early, Late and Delayed FRI Remains Unclear

This study does not address the presence, absence or maturity of bacterial biofilm with time from injury. Bacteria in biofilm exist within a polysaccharide matrix in a stationary phase of growth, therefore making them more difficult to treat [32]. Furthermore, models of biofilm formation suggest that bacteria in biofilm become less susceptible to antibiotics with time, and it is likely that this biofilm maturation takes place over a few days [33,34]. Mature biofilm is also seen by electron microscopy in mouse models of orthopaedic metalwork-associated infections at 14 days [35]. Therefore, by extension, the division of fracture-related infections into early (<2 weeks), delayed (3–10 weeks) and late (>10 weeks) based on the hypothesized presence of biofilm seems illogical, given that biofilm is most likely to have matured within the first 2 weeks of infection. 

### 3.4. Biofilm-Active Antibiotics and FRI Managed with DAIR

The role of biofilm-active antibiotics in treatment of FRI is also unclear. Assumptions are often extrapolated from data from PJI in which rifampicin has been shown to be most active against biofilm forming *S. aureus* [36,37]. Consequently, rifampicin is recommended in the latest antimicrobial consensus document for when FRI is managed with DAIR [5]. Indeed, Depypere et al. suggest that the rifampicin-resistant DTT pathogens cannot be successfully eradicated if the implant is retained, discouraging DAIR for patients where no anti-biofilm antibiotic option is available [5]. However, a recent randomized control trial and meta-analysis found no—or at best a minimal—advantage of including rifampicin in antimicrobial regimes for patients with *S. aureus* PJI managed with DAIR [38,39]. Furthermore, another study found no statistically significant difference in the outcome of patients with PJI caused by DTT pathogens, perhaps casting doubt on the assumptions underlying current clinical practice [16]. 

### 3.5. Limitations and Future Work

This is a multi-centre international study; however, the diversity of pathogens isolated is still only representative of those found at three European centres. Furthermore, differences in culture technique could inadvertently bias the pathogens identified at each centre, though each protocol used is considered valid. The small numbers of resistant pathogens isolated mean that further analysis of these pathogens with regard to time from injury in FRI is not possible. Additional analyses to search for microbiological associations with mechanism or location of injury (e.g., upper limb versus lower limb) may further guide empiric and culture-negative antimicrobial therapy. This study also highlights that further work is necessary to attempt to quantify the timescale of biofilm formation in clinical samples, and to address the role of biofilm-active antibiotics in management of FRI, particularly for patients with DTT pathogens. 

## 4. Materials and Methods

All patients with FRI, as defined by the confirmatory criteria of the FRI consensus definition, at three specialist centres between 2015–2019 and followed up for at least one year, were included. Patients were excluded if FRI affected the spine or skull. Data was collected by retrospective notes review, including patient demographics, characteristics of the injury, fracture and FRI surgery and the microorganisms isolated. Microbiological diagnosis was determined from culture of at least three deep tissue specimens, harvested during surgery. Specimens were taken with separate instruments, to avoid cross-contamination [23,40,41]. Culture results from superficial swabs or sinus tracts were considered insufficient, and not included in this study. 

At one centre, intraoperative samples were collected in sterile universal containers pre-filled with 3 mL saline and sterile glass beads (Equine and Ovine Laboratories). Samples were disrupted by vortexing at 40 Hertz for 15 s. Any metalwork removed was placed in a sonication container and covered at least 90% with sterile saline. Containers were vortexed for 30 s, sonicated in an ultrasound bath for 1 min and vortexed again for 30 s. An amount of 0.5 mL of each sample was inoculated into a BD BACTEC™ Lytic 10 Anaerobic/F bottle and 0.5 mL was inoculated into a BD BACTEC™ Lytic 10 Aerobic/F bottle. Culture bottles were incubated for up to 10 days at 37 °C and any that flagged positive were subcultured on agar [24,42]. 

At the second centre, intra-operative samples were cultured at 37 °C for 9–14 days on blood and chocolate agar under aerobic conditions (with 5% CO_2_) and on Brucella agar under anaerobic conditions. Tissue samples were also cultured in fastidious broth (Media products, Groningen, The Netherlands) and then subcultured onto blood and Brucella agar at 7 days, or earlier if deemed positive. Any metalwork was sonicated in sterile Ringers lactate at 40,000 Hz for 1 min. 0.1 mL of sonication fluid was plated onto blood agar plates, and 10 mL inoculated into BD BACTEC™ blood culture bottles and incubated for up to 9 days at 37 °C. Any that flagged positive were subcultured onto blood agar [43]. 

At the third centre, intra-operative bone samples were cultured in brain-heart infusion broth with added hemin and nicotinamide adenine dinucleotide and incubated for 7 days. Positive cultures were subcultured onto blood agar and chocolate agar and incubated aerobically (5% CO_2_), or blood agar and Brucella agar and incubated anaerobically. Tissue samples were homogenised using beads and then cultured in the same way as bone samples, as well as directly on blood agar, chocolate agar, McConkey agar and Brucella agar. Removed metalwork was submerged in sterile sodium chloride 0.9% and sonicated for 1 min at maximum power (Bandelin BactoSonic) before culture using blood and chocolate agar plates, Brucella agar and thioglycolate enrichment broth.

Importantly, it has been shown that the use of BACTEC bottles and cooked meat enrichment broth have similar sensitivity and specificity when detecting causative pathogens in PJIs [44]. Similarly, using BACTEC bottles, 95% of clinically relevant organisms were isolated within 3 days using aerobic cultures and 96% were isolated within 5 days using anaerobic cultures [42]. 

At all centres, microorganisms were identified by matrix-assisted laser desorption/ionization time-of-flight (MALDI-TOF) mass spectrometry (Bruker) or, in one case, via 16S PCR. Drug susceptibility testing was performed using the BD Phoenix system (BD Diagnostics) or manual methods including disc diffusion assays or antibiotic gradient strips, and comparison with EUCAST breakpoints (www.EUCAST.org, last accessed on 14 June 2022).

Microorganisms were considered to be causative pathogens if phenotypically indistinguishable isolates were found in two or more intra-operative samples or, in cases where the diagnosis of FRI was made on non-microbiological criteria, if a virulent pathogen was isolated in one sample (virulent pathogens were defined as *S. aureus*, *S. lugdunensis*, beta haemolytic streptococci, *Streptococcus milleri* group, *Enterococcus species*, *Enterobacteriaceae*, *Pseudomonas aeruginosa*, anaerobic Gram-negative bacilli and *Candida species*). In contrast, single isolates of *S. epidermidis* and *Corynebacterium species*, for example, were considered contaminants, and where this was the only positive microbiology, or where no pathogens were isolated, the case would be considered culture-negative. Cases were considered polymicrobial where two or more different pathogens were isolated. Isolates were classified as methicillin-resistant, rifampicin-resistant or ciprofloxacin-resistant using the standard susceptibility test methods described above. In addition, carbapenemase activity was confirmed using the MAST^®^ Indirect Carbapenemase Test (Mast Group, Liverpool, UK) followed by the NG Test^®^ CARBA 5 lateral flow assay (Hardy Diagnostics, USA). Extended spectrum beta lactamase activity was confirmed using Rosco Neo-Sensitabs™ (Rosco, Denmark). Patients identified as being managed by DAIR were analysed separately.

FRIs were classified according to time between initial injury and surgery for FRI as early (0–2 weeks), delayed (3–10 weeks) or late infections (>10 weeks). Use of this time interval was a pragmatic decision, as timing of symptom onset can be subject to recall bias whereas injury date is almost always known. All surgery was considered appropriate by consensus agreement between expert orthopaedic surgeons. Statistical analysis was performed using GraphPad Prism 9 (GraphPad Software, San Diego, CA, USA). Chi-squared tests were used for non-continuous variables. Continuous variables were analysed by Wilcoxon Signed Rank, after testing for normality using Shapiro–Wilk’s test. Post hoc testing was performed with Bonferroni correction. 

Ethical approval was granted by institutional clinical audit approval (OUH 2021/7566 and UMCU 20-004/C). A data sharing agreement was co-signed by all legal parties.

## 5. Conclusions

Our data demonstrate that decisions on FRI treatment should not assume microbiological differences related to time from injury. Similarly, time from injury should not be used to guide empiric antimicrobial therapy. When managing culture-negative infections, clinicians must consider antibiotic regimes active against the diversity of pathogens isolated in all culture-positive infections. PCR may be useful in culture-negative cases to identify the causative pathogens. Further work is required to examine the role of biofilm formation, biofilm-active antibiotics and antimicrobial resistance in FRI. However, currently, the microbiological relevance of classifying FRI by time from injury remains unclear.

## Figures and Tables

**Figure 1 antibiotics-11-00943-f001:**
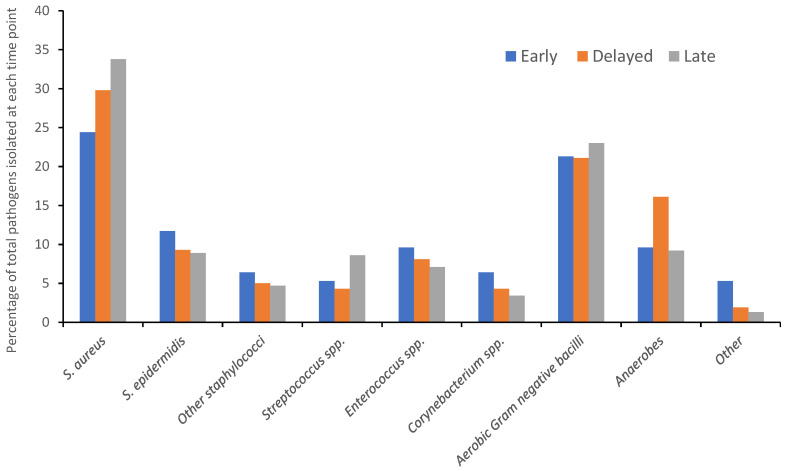
The distribution of pathogens isolated in early, delayed and late fracture-related infections.

**Table 1 antibiotics-11-00943-t001:** Patient Demographics and Microbiological Results by Time Since Injury.

	0–2 Weeks	3–10 Weeks	>10 Weeks	Whole Group	Significance
**Patient Demographics**					
Number of cases	51	82	300	433	
Age (median, years; IQR)	50; 32–60	52; 36–63	50; 37–62	51; 36–62	*p* = 0.85
Sex (% male)	67	67	76	70	*p* = 0.12
BMI (median; IQR)	23.6 *; 22.8–24.9	25.4; 23.0–29.6	27.8 *; 23.6–31.6	27.3; 23.4–31.0	* *p* = 0.004
Time since injury (median, weeks; IQR)	2; 1–2	5; 4–8	112; 40–737	44; 6–342	
**Bone Involved**					
Tibia/fibula	24 (47%)	47 (57%)	166 (55%)	237	*p* < 0.001
Femur	10 (20%)	8 (10%)	76 (25%)	94	
Upper limb	5 (10%)	8 (10%)	39 (13%)	52	
Pelvis	8 (16%)	11 (13%)	7 (2%)	26	*p* = 0.17 (tibia/fibula, femur, upper limb only)
Foot	4 (8%)	7 (9%)	8 (3%)	19	
Other	0	1 (1%)	4 (1%)	5	
**Culture Type**					
Culture-negative	2 (4%) ***	6 (7%)	72 (24%) ***	80 (19%)	*** *p* < 0.0001
Monomicrobial	19 (37%)	30 (37%)	150 (50%)	199 (46%)	
Polymicrobial	30 (59%) **	46 (56%) ***	78 (26%) **^,^***	154 (36%)	** *p* = 0.0004, *** *p* < 0.0001
**Species Isolated**					
*S. aureus*	23 (24%)	48 (30%)	129 (34%)	200 (31%)	*p* = 0.20
*S. epidermidis*	11 (12%)	15 (9%)	34 (9%)	60 (9%)	
Other staphylococci	6 (6%)	8 (5%)	18 (5%)	32 (5%)	
*Streptococcus* spp.	5 (5%)	7 (4%)	33 (9%)	45 (7%)	
*Enterococcus* spp.	9 (10%)	13 (8%)	27 (7%)	49 (8%)	
*Corynebacterium* spp.	6 (6%)	7 (4%)	13 (3%)	26 (4%)	
Aerobic Gram-neg. bacilli	20 (21%)	34 (21%)	88 (23%)	142 (22%)	
Anaerobes	9 (10%)	26 (16%)	35 (9%)	70 (11%)	
Other	5 (5%)	3 (2%)	5 (1%)	13 (2%)	
Total	94	161	382	637	

* Significant difference between marked groups *p* < 0.05; ** significant difference between marked groups *p* < 0.001; *** significant difference between marked groups *p* < 0.0001; BMI: body mass index; DAIR: debridement, antimicrobial therapy and implant retention; Gram-neg.: Gram-negative; IQR: interquartile range; spp.: species.

**Table 2 antibiotics-11-00943-t002:** Subgroup Analysis of Patients Managed with DAIR by Time Since Injury.

	0–2 Weeks	3–10 Weeks	>10 Weeks	Whole Group	Significance
**Patient Demographics**					
Number of cases	45	62	33	140	
Age (median, years; IQR)	50; 32–60	53; 37–63	55; 36–64	53; 35–63	*p* = 0.63
Sex (% male)	64	63	73	66	*p* = 0.62
BMI (median; IQR)	23.5 *; 22.8–24.3	25.5; 23.4–29.9	30.7 *; 26.2–35.5	25.7; 23.1–31.2	* *p* = 0.027
Time since injury (median, weeks; IQR)	2; 1–2	5; 4–6	45; 28–97	4; 2–8	
**Bone Involved**					
Tibia/fibula	21 (47%)	34 (55%)	15 (45%)	70	*p* = 0.26
Femur	9 (20%)	7 (11%)	10 (30%)	26	
Upper limb	5 (11%)	5 (8%)	4 (12%)	14	
Pelvis	8 (18%)	10 (16%)	2 (6%)	20	
Foot	2 (4%)	6 (10%)	1 (3%)	9	
Other	0 (0%)	0 (0%)	1 (3%)	1	
**Culture Type**					
Culture-negative	2 (4%)	3 (5%)	2 (6%)	7 (5%)	*p* = 0.52
Monomicrobial	18 (40%)	23 (37%)	18 (55%)	59 (42%)	
Polymicrobial	25 (56%)	36 (58%)	13 (39%)	74 (53%)	
**Species Isolated**					
*S. aureus*	20 (24%)	40 (33%)	21 (35%)	81 (30%)	*p* = 0.56
*S. epidermidis*	8 (10%)	12 (10%)	8 (13%)	28 (11%)	
Other staphylococci	5 (6%)	6 (5%)	4 (7%)	15 (6%)	
*Streptococcus* spp.	4 (5%)	6 (5%)	6 (10%)	16 (6%)	
*Enterococcus* spp.	8 (10%)	8 (7%)	3 (5%)	19 (7%)	
*Corynebacterium* spp.	6 (7%)	5 (4%)	3 (5%)	14 (5%)	
Aerobic Gram-neg. bacilli	17 (21%)	19 (16%)	5 (8%)	41 (16%)	
Anaerobes	10 (12%)	23 (19%)	10 (17%)	43 (16%)	
Other	4 (5%)	3 (3%)	0	7 (3%)	
Total	82	122	60	264	

* Significant difference between marked groups *p* < 0.05; BMI: body mass index; DAIR: debridement, antimicrobial therapy and implant retention; Gram neg.: Gram-negative; IQR: interquartile range; spp.: species.

## Data Availability

Data is available from the corresponding author on request.

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
