# Peer review of "Causative Pathogens Do Not Differ between Early, Delayed or Late Fracture-Related Infections"

_antibiotics, 2022, doi:10.3390/antibiotics11070943_

Round 1
Reviewer 1 Report
The manuscript "Causative pathogens do not differ between early, delayed or late fracture-related infections" is, in my opinion, well-written and quite interesting. However, I have some minor comments:
1. In the Abstract there are some linguistic mistakes:
- line 20: "...is classically defined as...". I suggest changing it into "...classically divided into three stages/phases..." or something of the similar meaning.
-line 23-24: "...pathogens isolated in FRI...". Shouldn't be "isolated from"?
-line 26: "analyzed on patient demographic...". Please try "analyzed considering/regarding:...".
2. In the Introduction (line 47) I suggest changing "numerous publications have sought to clarify" to "numerous researchers intended/tried/attempted to clarify".
3. In the Materials and Methods (line 339-340) I suggest providing additional information about "standard laboratory resistance tests" that were mentioned in the paragraphed, but not described in detail.
4. Please add some information in the Conclusions section because it seems to be too short in the present form.
Author Response
Response to Reviewer 1 Comments
Point 1:
In the Abstract there are some linguistic mistakes:
- line 20: "...is classically defined as...". I suggest changing it into "...classically divided into three stages/phases..." or something of the similar meaning.
-line 23-24: "...pathogens isolated in FRI...". Shouldn't be "isolated from"?
-line 26: "analyzed on patient demographic...". Please try "analyzed considering/regarding:...".
Response 1:
I have made the changes as suggested above.
Point 2:
In the Introduction (line 47) I suggest changing "numerous publications have sought to clarify" to "numerous researchers intended/tried/attempted to clarify".
Response 2:
I have made the change as suggested above (line 55).
Point 3:
In the Materials and Methods (line 339-340) I suggest providing additional information about "standard laboratory resistance tests" that were mentioned in the paragraphed, but not described in detail.
Response 3:
At line 378-9 I have added the comment ‘including disc diffusion assays or antibiotic gradient strips and comparison with EUCAST breakpoints (www.EUCAST.org).’ and modified line 390-395 to read ‘Isolates were classified as methicillin resistant, rifampicin resistant or ciprofloxacin resistant using the standard susceptibility test methods described above. In addition, carbapenemase activity was confirmed using the MAST® Indirect Cabapenemase Test (Mast Group, Liverpool UK) followed by the NG Test® CARBA 5 lateral flow assay (Hardy Diagnostics, USA). Extended spectrum beta lactamase activity was confirmed using Rosco Neo-Sensitabs™ (Rosco, Denmark).
Point 4:
Please add some information in the Conclusions section because it seems to be too short in the present form.
Response 4:
This section was deliberately brief given the instructions to authors on the manuscript submission template ‘Conclusions: This section is not mandatory but can be added to the manuscript if the discussion is unusually long or complex.’
However, given the comment above I have amended the conclusion to read ‘Our data demonstrate that decisions on FRI treatment should not assume microbiological differences related to time from injury. Similarly, time from injury should not be used to guide empiric antimicrobial therapy. When managing culture negative infections clinicians must consider antibiotic regimes active against the diversity of pathogens isolated in all culture positive infections. PCR may be useful in culture negative cases to identify the causative pathogens. Further work is required to examine the role of biofilm formation, biofilm active antibiotics and antimicrobial resistance in FRI. However, currently, the microbiological relevance of classifying FRI by time from injury remains unclear.’ (Lines 419-426).
Reviewer 2 Report
Authors wrote an interesting paper on fracture related infection. The research question in important and the paper result well write.
Below my minor suggestion that hope can improve the paper.
Introduction: add data on antimicrobial resistance wordwilde
Methods and results clear
Discussion: discuss better the long acting gram + MRSA (ex Dalbavancin and other new possible therapy) and the role and new challenge on infection difficult to treat
Furthermore, add also the role of medical education to increase awareness in correct use of antibiotic (see Italian young doctors' knowledge, attitudes and practices on antibiotic use and resistance: A national cross-sectional survey. J Glob Antimicrob Resist. 2020 Dec;23:167-173)
Discuss better also the role of the cost of infection on health system and the new approach on biolfilm ant its role
Conclusion: give some proposal that came from your interesting paper
Author Response
Response to Reviewer 2 Comments
Point 1:
Introduction: add data on antimicrobial resistance worldwide
Response 1:
I have added ‘Indeed, increasing antimicrobial resistance worldwide [14] is also reflected in cases of osteomyelitis over time [15] although data regarding the role of antimicrobial resistance in FRI is currently lacking’ to lines 80-82.
Point 2:
Discussion: discuss better the long acting gram + MRSA (ex Dalbavancin and other new possible therapy) and the role and new challenge on infection difficult to treat
Response 2:
I have added ‘Although increasing antimicrobial resistance has been observed in osteomyelitis over time and presents treatment challenges [15], such trends may vary depending on trends in antimicrobial use [22] and highlight the need for antimicrobial stewardship in orthopaedic infections [23]’ to lines 278-281.
Although I can add comments regarding treatment options for MRSA+ infections, including dalbavancin I feel this is beyond the remit of this paper. I have added the comment that antimicrobial resistance provides treatment challenges and I feel this is sufficient.
Point 3:
Add also the role of medical education to increase awareness in correct use of antibiotic (see Italian young doctors' knowledge, attitudes and practices on antibiotic use and resistance: A national cross-sectional survey. J Glob Antimicrob Resist. 2020 Dec;23:167-173)
Response 3:
In addition to the response to point 2, I had added the following ‘Educating prescribers on the consequences of inappropriate antimicrobial use must therefore be a priority [24]’ to lines 281-282 including the reference suggested.
Point 4:
Discuss better also the role of the cost of infection on health system and the new approach on biofilm and its role
Response 4:
With regard to biofilm and its treatment I have added the following section to the discussion:
‘3.4 Biofilm-active antibiotics and FRI managed with DAIR
The role of biofilm active antibiotics in treatment of FRI is also unclear. Assumptions are often extrapolated from data from PJI where rifampicin has been shown to be most active against biofilm active S. aureus [36, 37]. Consequently, rifampicin is recommended in the latest antimicrobial consensus document for when FRI is managed is DAIR [5]. Indeed, Depypere et al. suggest that the rifampicin resistant DTT pathogens cannot be successfully eradicated if the implant is retained, discouraging DAIR for patients where no anti-biofilm antibiotic option is available [5]. However, a recent randomized control trial and meta-analysis found no, or at best a minimal, advantage of including rifampicin in antimicrobial regimes for patients with S. aureus PJI managed with DAIR [38, 39]. Furthermore, another study found no statistically significant difference in the outcome of patients with PJI caused by DTT pathogens, perhaps casting doubt on the assumptions underlying current clinical practice [16].’ (lines 306-318).
With regard to cost of infection, in the introduction I have added ‘This complication is associated with both increased mortality and morbity as well as socioeconomic cost [2], making the finding of effective treatment strategies a priority.’ (lines 51-53).
Point 5:
Conclusion: give some proposal that came from your interesting paper
Response 5:
I have amended the conclusion to read:
‘Our data demonstrate that decisions on FRI treatment should not assume microbiological differences related to time from injury. Similarly, time from injury should not be used to guide empiric antimicrobial therapy. When managing culture negative infections clinicians must consider antibiotic regimes active against the diversity of pathogens isolated in all culture positive infections. PCR may be useful in culture negative cases to identify the causative pathogens. Further work is required to examine the role of biofilm formation, biofilm active antibiotics and antimicrobial resistance in FRI. However, currently, the microbiological relevance of classifying FRI by time from injury remains unclear.’ (Lines 419-426).

Reviewer 3 Report
review" Causative pathogens do not differ between early, delayed or late fracture-related infections "
Based on postulated variations in the causative microorganisms and biofilm formation, fracture-related infection (FRI) is often classified as early (0–2 weeks), delayed (3–10 weeks), or late (>10 weeks). The present paper analyzes the treatment plans frequently correspond to this classification, with early FRI being given preference for debridement, antimicrobial therapy, and implant retention (DAIR). The topic is very interesting, although the author may have to add and reorganize the article before publication.
As a first point, I would like to point out that the introduction section is too small; I would suggest that the authors add a few more references.
In my humble opinion, the methods section should be after the introduction and before the result section. It may help the readers to know the methodology in order to evaluate the results.
Some of the tables have incorrect legends. It should be corrected.
The discussion section and the conclusion section are written poorly. The discussion section is very important to compare the published results by other colleagues with their findings. This section needs to be improved. Additionally, the conclusion section needs improvement.
Author Response
Response to Reviewer 3 Comments
Point 1:
The introduction section is too small; I would suggest that the authors add a few more references.
Response 1:
I have contextualised the clinical importance of fracture related infection with two additional references (lines 50-53) and expanded on some other considerations with regard to treatment choice with one additional reference (lines 66-68). As per Reviewer 2, I have added comments regarding antimicrobial resistance in orthopaedic infections and two additional references (lines 80-82).
Point 2:
In my humble opinion, the methods section should be after the introduction and before the result section. It may help the readers to know the methodology in order to evaluate the results.
Response 2:
The methods section is Section 4 (between 3. Discussion and 5. Conclusion) as per the template for authors for submission to this journal. Currently, I have left it as Section 4.
Point 3:
Some of the tables have incorrect legends. It should be corrected.
Response 3:
For Table 2 I have amended the legend to ‘Table 2. Subgroup Analysis of Patients Managed with DAIR by Time Since Injury’ (line 114). I am unclear from the reviewer and my own close inspection what else is being referred to by this point. Please clarify if incorrect legends remain and I will happily amend.
Point 4:
The discussion section is very important to compare the published results by other colleagues with their findings. This section needs to be improved.
Response 4:
I have more explicitly elaborated on our findings in comparison with publish studies:
- Lines 233-236 ‘For example, in a recent review Depypere et al. found that S. aureus was isolated in 30-42% of FRI compared to 31% in this study [10]. Our findings that infections were most common in males and in lower extremity fractures correlates with known risk factors for FRI [20].’
- Lines 241-247 ‘Walter et al. find only 9.4% of FRI cases are culture negative compared to 19% overall in our study, but all of these culture negative infections were found in delayed or late FRI [19]. Our culture negative rate is higher than other published reports [21], however this may be due to the high percentage of late infections in our cohort. Of note, a recent study of PJI defined according to consensus definitions found zero cases of culture negative infections in acute cases versus 70 (4.7%) in chronic infections which reflects our findings in FRI [22].’
- Line 259-260 ‘Polymerase chain reaction (PCR) has been shown to be useful to isolate pathogens in culture negative PJI though studies looking at its utility in FRI are still lacking [28].
Point 5:
The conclusion section needs improvement.
Response 5:
I have amended the conclusion to read:
‘Our data demonstrate that decisions on FRI treatment should not assume microbiological differences related to time from injury. Similarly, time from injury should not be used to guide empiric antimicrobial therapy. When managing culture negative infections clinicians must consider antibiotic regimes active against the diversity of pathogens isolated in all culture positive infections. PCR may be useful in culture negative cases to identify the causative pathogens. Further work is required to examine the role of biofilm formation, biofilm active antibiotics and antimicrobial resistance in FRI. However, currently, the microbiological relevance of classifying FRI by time from injury remains unclear.’ (Lines 419-426).
Round 2
Reviewer 3 Report
There is a significant improvement in the quality of the updated version of the paper.
The authors responded to all comments. In its current form, the paper can be accepted.